# Learning Multi-Objective Curricula for Robotic Policy Learning

**Jikun Kang**
McGill University
Mila - Québec AI Institute
jikun.kang@mail.mcgill.ca

**Miao Liu**
IBM Research
Miao.Liu1@ibm.com

**Abhinav Gupta**
Université de Montréal
Mila - Québec AI Institute
abhinavg@nyu.edu

**Christopher Pal**
Polytechnique Montréal
Mila - Québec AI Institute
christopher.pal@polymtl.ca

**Xue Liu**
McGill University
Mila - Québec AI Institute
xueliu@cs.mcgill.ca

**Jie Fu**[*]
Beijing Academy of AI
fujie@baai.ac.cn

**Abstract:** Various automatic curriculum learning (ACL) methods have been proposed to improve the sample efficiency and final performance of robots' policies learning. They are designed to control how a robotic agent collects data, which is inspired by how humans gradually adapt their learning processes to their capabilities. In this paper, we propose a unified automatic curriculum learning framework to create multi-objective but coherent curricula that are generated by a set of parametric curriculum modules. Each curriculum module is instantiated as a neural network and is responsible for generating a particular curriculum. In order to coordinate those potentially conflicting modules in a unified parameter space, we propose a multi-task hyper-net learning framework that uses a single hyper-net to parameterize all those curriculum modules. We evaluate our method on a series of robotic manipulation tasks and demonstrate its superiority over other state-of-the-art ACL methods in terms of sample efficiency and final performance. Our code is available at https://github.com/luciferkonn/MOC_CoRL22.

**Keywords:** ACL, Hyper-net, Multi-objective Curricula

## 1 Introduction

The concept that humans frequently organize their learning into a curriculum of interdependent processes according to their capabilities was first introduced to machine learning in [1]. Over time, curriculum learning has become more widely used in machine learning to control the stream of examples provided to training algorithms [2], to adapt model capacity [3], and to organize exploration [4]. Automatic curriculum learning (ACL) for deep reinforcement learning (DRL) [5] has recently emerged as a promising tool to learn how to adapt a robot's learning tasks to its capabilities during training. ACL can be applied to robotics' policies learning in various ways, including adapting initial states [6], shaping reward functions [7], generating goals [8]. More broadly, curriculum learning can also be used to modify the environment itself. For example, it can add new entities to the world or change the behaviors of other agents [9].

Oftentimes, only a single ACL paradigm (*e.g.*, generating subgoals) is considered. It remains an open question whether different paradigms are complementary to each other and if yes, how to combine them in a more effective manner similar to how the "rainbow" approach of [10] has greatly improved DRL performance in Atari games. Multi-task learning is notoriously difficult, and Yu et al. [11] hypothesize that the optimization difficulties might be due to the gradients from different tasks

---

[*]Corresponding Author

6th Conference on Robot Learning (CoRL 2022), Auckland, New Zealand.

conflicting with each other thus hurting the learning process. In this work, we propose a multi-task bilevel learning framework for more effective multi-objective curricula robotics' policy learning. Concretely, inspired by neural modular systems [12] and multi-task RL [13], we utilize a set of neural modules and train each of them to output a sequence of tasks with different desired goals, rewards, initial states, etc. To coordinate potentially conflicting gradients from modules in a unified parameter space, we use a single hyper-net [14] to parameterize neural modules so that these modules generate a diverse and cooperative set of curricula. *Multi-task learning provides a natural curriculum for the hyper-net itself* since learning easier curriculum modules can be beneficial for learning more difficult curriculum modules with parameters generated by the hyper-net.

Furthermore, existing ACL methods usually rely on manually-designed paradigms of which the target and mechanism have to be clearly defined and it is therefore challenging to create a very diverse set of curriculum paradigms. Consider goal-based ACL for example, where the algorithm is tasked with learning how to rank goals to form the curriculum [15]. Many of these curriculum paradigms are based on simple intuitions that are inspired by learning in humans, but they usually take too simple forms (*e.g.*, generating subgoals) to apply to neural models. Instead, we propose to augment the hand-designed curricula introduced above with an *abstract* curriculum of which paradigm is learned from scratch. More concretely, we take the idea from memory-augmented meta-DRL [16] and equip the hyper-net with a non-parametric memory module, which is also directly connected to the DRL agent. The hyper-net can write entries to and update items in the memory, through which the DRL agent can interact with the environment under the guidance of the abstract curriculum maintained in the memory. The *write-only* permission given to the hyper-net over the memory is distinct from the common use of memory modules in meta-DRL literature, where the memories are both readable and writable. We point out that the hyper-net is instantiated as a recurrent neural network [17] which has its internal memory mechanism and thus a write-only extra memory module is enough. Another key perspective is that *such a write-only memory module suffices to capture the essence of many curriculum paradigms*. For instance, the subgoal-based curriculum can take the form of a sequence of coordinates in a game which can be easily generated as a hyper-net and stored in the memory module. The contributions of this work are as follows:

1. We introduce multi-objective curricula learning approach for improving the sample efficiency of solving challenging deep reinforcement learning tasks, which is an important problem that has not been adequately addressed before.

2. We further propose a unified automatic curriculum learning framework to create multi-objective but coherent curricula that are generated by a set of parametric curriculum modules. Each curriculum module is instantiated as a neural network and is responsible for generating a particular curriculum. To coordinate those potentially conflicting modules in unified parameter space, we propose a multi-task hyper-net learning framework that uses a single hyper-net to **parameterize** all those curriculum modules.

## 2 Related Work

**Curriculum learning.** Automatic curriculum learning (ACL) for deep reinforcement learning (DRL) [18, 5, 19, 20, 21] has recently emerged as a promising tool to learn how to adapt an agent's learning tasks based on its capacity during training. ACL [22] can be applied to DRL in a variety of ways, including adapting initial states [6, 23], shaping reward functions [24, 25], or generating goals [8, 15, 26, 27]. In a closely related work [28], a series of related environments of increasing difficulty have been created to form curricula. There are other works related to curriculum reinforcement learning (CRL).

**Multi-task and neural modules.** Learning with multiple objectives is shown to be beneficial in DRL tasks [29, 30, 31, 32]. Sharing parameters across tasks [33, 34, 35] usually results in conflicting gradients from different tasks. One way to mitigate this is to explicitly model the similarity between gradients obtained from different tasks [11, 36, 37, 38, 39, 40, 41]. On the other hand, researchers propose to utilize different modules for different tasks, thus reducing the interference of gradients

from different tasks [42, 43, 44, 45, 46, 47, 48]. Most of these methods rely on pre-defined modules that make them not flexible in practice. One exception is [12], which utilizes soft combinations of neural modules for multi-task robotics manipulation. However, there is still redundancy in the modules in [12], and those modules cannot be modified during inference. Instead, we use a hyper-net to dynamically update complementary modules on the fly conditional on the environments.

**Memory-augmented meta DRL.** Our approach is also related to episodic memory-based meta DRL [49, 16, 50, 51]. Different from memory-augmented meta DRL methods, the DRL agent in our case is not allowed to modify the memory. This design prevents the agent from writing its information into the memory module, which can interfere with the generated curriculum. Note that it is straightforward to augment the DRL agent with both readable and writable neural memory just like [16, 49], which is different from our read-only memory module designed for ACL.

**Dynamic neural networks.** Dynamic neural networks [52] can change their structures or parameters based on different environments. Dynamic filter networks [53] and hyper-nets [14] can both generate parameters.

Our proposed framework generalizes previous work by unifying the aforementioned key concepts with a focus on automatically learning multi-objective curricula from scratch for DRL-based robot learning.

## 3   Preliminaries

**Reinforcement learning (RL)** is used to train an agent policy with the goal of maximizing the (discounted) cumulative rewards through trial and error. A basic RL setting is modelled as a Markov decision process (MDP) with the following elements: $\mathcal{S}$ is the set of environment states. In this paper, goal $g \in \mathcal{G}$ corresponds to a set of states $S^g \subset S$. The goal is considered to be achieved when the agent is in any state $s_t \in S^g$; $\mathcal{A}$ is the set of actions; $\delta$ is the state transition probability function, where $\delta(s^{t+1}|s^t, a^t)$ maps a state–action pair at time-step $t$ to a probability distribution over states at time $t + 1$; $R$ is the immediate reward after a transition from $s$ to $s'$; $\pi(\cdot; \phi_\pi)$ is the policy function parameterized by $\phi_\pi$, and $\pi(a|s; \phi_\pi)$ denotes the probability of choosing action $a$ given an observation $s$.

**Automatic curriculum learning (ACL)** is a learning paradigm where an agent is trained iteratively following a curriculum to ease learning and exploration in a multi-task problem. Since it is not feasible to manually design a curriculum for each task, recent work has proposed to create an implicit curriculum directly from the task objective. Concretely, it aims to maximize a metric $P$ computed over a set of target tasks $T \sim \mathcal{T}_{target}$ after some episodes $t'$. Following the notation in [5], the objective is set to: $\max_{\mathcal{D}} \int_{T \sim \mathcal{T}_{target}} P_T^{t'} dT$, where $\mathcal{D} : \mathcal{H} \to \mathcal{T}_{target}$ is a task selection function. The input $\mathcal{H}$ can be consist of any information about past interactions. For example, in our experiments, the history consists of the last state of the episode. , and the output of $\mathcal{D}$ is a sequence of generated episodes, which contain different desired goals, initial states, rewards, etc.

**Hyper-networks** were proposed in [14] where one network (hyper-net) is used to generate the weights of another network. All the parameters of both networks are trained end-to-end using backpropagation. We follow the notation in [54] and suppose that we aim to model a target function $y : \mathcal{X} \times \mathcal{I} \to \mathbb{R}$, where $x \in \mathcal{X}$ is independent of the task and $I \in \mathcal{I}$ depends on the task. A **base** neural network $f_b(x; f_h(I; \theta_h))$ can be seen as a composite function, where $f_b : \mathcal{X} \to \mathbb{R}$ and $f_h : \mathcal{I} \to \Theta_b$. Conditioned on the task information $I$, the small **hyper**-net $f_h(I; \theta_h)$ generates the parameters $\theta_b$ of base-net $f_b$. Note that $\theta_b$ is never updated using loss gradients directly.

## 4   Learning Multi-Objective Curricula

We use a single hyper-net to dynamically parameterize all the curriculum modules over time and modify the memory module shared with the DRL agent. We call this framework a Multi-Objective

Curricula (MOC). This novel design encourages different curriculum modules to merge and exchange information through the shared hyper-net.

Following the design of hyper-networks with recurrence [14], this hyper-net is instantiated as a recurrent neural network (RNN), which we refer to as the **Hyper-RNN**, denoted as $f_h(I; \theta_h)$, in the rest of this paper to emphasize its dynamic nature. Additionally, the Hyper-RNN can be viewed as a configurator for other modules as suggested in [55]. Our motivation for the adoption of an RNN design is its capability for producing a distinct set of curricula for every episode, which strikes a better trade-off between the number of model parameters and its expressiveness. On the other hand, each manually designed curriculum module is also instantiated as an RNN, which is referred as a **Base-RNN** $f_b(x; \theta_b)$ parameterized by $\theta_b = f_h(I; \theta_h)$. Each Base-RNN is responsible for producing a specific curriculum, *e.g.*, a series of sub-goals.

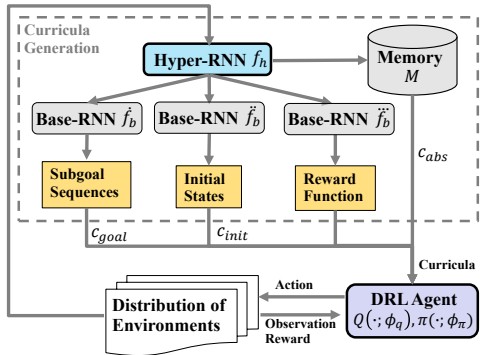

Figure 1: Illustration of MOC-DRL with two loops. Curricula generation corresponds to the outer-level loop. The DRL agent interacts with the environment in the inner-level loop.

**Algorithm 1:** Multi-Objective Curricula Deep Reinforcement Learning (MOC-DRL).

1 **for** *Episode $t'$ in 1 to $T_{outer}$* **do**
2    · Sample a new environment from the distribution of environments;
3    · Hyper-RNN generates parameters for each curriculum module;
4    **for** *Base-RNN in 1 to 3* **do**
5       · Generate a curriculum;
6    · Hyper-RNN updates the abstract curriculum in the memory;
7    **for** *Training step $t$ in 1 to $T_{inner}$* **do**
8       · DRL agent reads memory;
9       · Train DRL agent following curricula;
10    · Update Hyper-RNN based on outer-level objective;

The architecture of MOC-DRL is depicted in Fig. 1, and its corresponding pseudo-code is given in Alg. 1. We formulate the training procedure as a bilevel optimization problem [56] where we minimize an outer-level objective that depends on the solution of the inner-level tasks.

In our case, the *outer*-level optimization comes from the curriculum generation loop where each step is an episode denoted as $t'$. On the other hand, the *inner*-level optimization involves a common DRL agent training loop on the interactions between the environment and the DRL agent, where each time-step at this level is denoted as $t$. We defer the discussion on the details to Sec. 4.3.

**Inputs**, $I$, of the Hyper-RNN, $f_h$, consist of: (1) the final state of the last episode, and (2) role identifier for each curriculum module (*e.g.*, for initial states generation) represented as a one-hot encoding. Ideally, we expect each Base-RNN to have its own particular role, which is specific to each curriculum. When generating the parameters for each Base-RNN, we additionally feed the role identifier representation to the Hyper-RNN.

**Outputs** of the Hyper-RNN at episode $t'$ include: (1) parameters $\theta_b^{t'}$ for each Base-RNN, and (2) the abstract curriculum, $\mathbf{h}_h^{t'}$, maintained in the memory module. Here $\mathbf{h}_h^{t'}$ corresponds to the hidden states of the Hyper-RNN such that $[\theta_b^{t'}, \mathbf{h}_h^{t'}] = f_h(I^{t'}; \theta_h)$.

## 4.1 Manually Designed Curricula

In Sec. 4.1, we describe the details of generating manually designed curricula while the process of updating the abstract curriculum is described in Sec. 4.2. We describe how to train them in Sec. 4.3.

In this work, we use three curriculum modules responsible for generating pre-defined curricula [5]: *initial state generator, sub-goal state generator,* and *reward shaping generator.* Our approach can be easily extended to include other forms of curricula (*e.g.*, selecting environments from a discrete set [57]) by adding another curriculum generator to the shared hyper-net. These Base-RNNs

simultaneously output the actual curricula for the DRL agent in a synergistic manner. It should be noted that these Base-RNNs are not directly updated by loss gradients, as their *pseudo-parameters* are generated by the Hyper-RNN.

**Generating subgoal state $g^t$ as curriculum $\mathbf{c}_{goal}$ with Base-RNN $\dot{f}_b$.** As one popular choice in ACL for DRL, the subgoals can be selected from discrete sets [8] or a continuous goal space [15]. A suitable subgoal state $g^t$ can ease the learning procedures by guiding the agent on how to achieve subgoals step by step and ultimately solving the final task.

To incorporate the subgoal state in the overall computation graph, in this paper, we adopt the idea from universal value functions [58] and modify the action-value function, $Q(\cdot; \phi_q)$, to combine the generated subgoal state with other information $Q := Q(s^t, a^t, g^t; \phi_q) = Q(s^t, a^t, \mathbf{c}_{goal}; \phi_q)$, where $s^t$ is the state, $a^t$ is the action, and $g^t$ is the generated subgoal state. The loss is defined as $\mathcal{J}_{goal} = \mathbb{E}_{(s^t, a^t, r^t, s^{t+1}, g^t) \sim H_{buf}}[(Q(s^t, a^t, \mathbf{c}_{goal}; \phi_q) - \dot{y})^2]$, where $\dot{y}$ is the one-step look-ahead:

$$\dot{y} = r^t + \lambda \mathbb{E}_{a^{t+1} \sim \pi_\theta(s^{t+1})}[Q(s^t, a^t, \mathbf{c}_{goal}; \phi_q) \quad - \log(\pi(a^{t+1}|s^{t+1}; \phi_\pi))], \tag{1}$$

$H_{buf}$ is the replay buffer and $\lambda$ is the discount factor.

**Generating initial state $\mathbf{s}_0$ as curriculum $\mathbf{c}_{init}$ with Base-RNN $\ddot{f}_b$.** Intuitively, if the starting state $\mathbf{s}_0$ for the agent is close to the end-goal state, the training would become easier, which forms a natural curriculum for training tasks whose difficulty depends on a proper distance between the initial state and the end-goal state. This method has been shown effective in control tasks with sparse rewards [6, 59]. To simplify implementation, even though we only need a single initial state $\mathbf{s}_0$ which is independent of time, we still use a Base-RNN, $\ddot{f}_b$, to output it.

The loss for this module is: $\mathcal{J}_{init} = \mathbb{E}_{(s^t, a^t) \sim H_{buf}}[(Q(s^t, a^t, c_{init}; \phi_q) - \dot{y})^2]$, where $\dot{y}$ is defined in Eqn. 1.

**Generating potential-based shaping function as curriculum $\mathbf{c}_{rew}$ with Base-RNN $\dddot{f}_b$.** Motivated by the success of using reward shaping for scaling RL methods to handle complex domains [60], we introduce reward shaping as the third manually selected curriculum. The reward shaping function can take the form of: $\dddot{f'}_b(s^t, a^t, s^{t+1}) = \mu \cdot \dddot{f}_b(s^{t+1}) - \dddot{f}_b(s^t)$, where $\mu$ is a hyper-parameter and $\dddot{f}_b()$ is base-RNN that maps the current state with a reward. In this paper, we add the shaping reward $\dddot{f'}_b(s^t, a^t, s^{t+1})$ to the original environment reward $r$. We further normalize the shaping reward between 0 and 1 to deal with wide ranges.

Following the optimal policy invariant theorem [60], we modify the look-ahead function: $\dddot{y} = r^t + \dddot{f}_b(s^t, a^t, s^{t+1}) + \lambda \mathbb{E}_{a^{t+1} \sim \pi_\theta(s^{t+1})}[Q(s^t, a^t, \mathbf{c}_{rew}; \phi_q) - \log(\pi(a^{t+1}|s^{t+1}; \phi_\pi))]$. Thus the loss is defined as: $\mathcal{J}_{reward} = \mathbb{E}_{s^t, a^t, s^{t+1}, a^{t+1} \sim H_{buf}}[(Q(s^t, a^t, \mathbf{c}_{rew}; \phi_q) - \dddot{y})^2]$.

### 4.2 Abstract Curriculum with Memory Mechanism

Although the aforementioned hand-designed curricula are generic enough to be applied in any environment/task, it is still limited by the number of such predefined curricula. It is reasonable to conjecture that there exist other curriculum paradigms, which might be difficult to hand-design based on human intuition. As a result, instead of solely asking the hyper-net to generate human-engineered curricula, we equip the hyper-nets with an external memory, in which the hyper-nets could read and update the memory's entries. The Hyper-RNN together with the memory can also be seen as an instantiation of shared workspaces [61] for different curricula modules, in which those modules exchange information.

By design, the content in the memory can serve as abstract curricula for the DRL agent, which is generated and adapted according to the task distribution and the agent's dynamic capacity during training. Even though there is no constraint on how exactly the hyper-net learns to use the memory, we observe that (see Sec. 5.2): 1) The hyper-net can receive reliable training signals from the manually designed curriculum learning objectives[2]; 2) Using the memory module alone would result in unstable

---
[2]To some extent, tasking the hyper-net to train manually designed curriculum modules can be seen as a curriculum itself for training the abstract curriculum memory module.

training; 3) Utilizing both the memory and manual curricula achieves the best performance and stable training. Thus, training this memory module with other manually designed curriculum modules contributes to the shaping of the content that can be stored in the memory and is beneficial for overall performance.

Specifically, external memory is updated by the Hyper-RNN. To capture the latent curriculum information, we design a neural memory mechanism similar to [62]. The form of memory is defined as a matrix $M$. At each episode $t'$, the Hyper-RNN emits two vectors $\mathbf{m}_e^{t'}$, and $\mathbf{m}_a^{t'}$ as $[\mathbf{m}_e^{t'}, \mathbf{m}_a^{t'}]^T = [\sigma, \texttt{tanh}]^T(\mathbf{W}_h^{t'}\mathbf{h}_h^{t'})$: where $\mathbf{W}_h^{t'}$ is the weight matrix of Hyper-RNN to transform its internal state $\mathbf{h}_h^{t'}$ and ${}^\mathsf{t}[\cdot]$ denotes matrix transpose. Note that $\mathbf{W}_h$ are part of the Hyper-LSTM parameters $\theta_h$.

The Hyper-RNN *writes* the abstract curriculum into the memory, and the DRL agent can *read* the abstract curriculum information freely.

**Reading.** The DRL agent can read the abstract curriculum $\mathbf{c}_{abs}$ from the memory $\mathbf{M}$. The read operation is defined as: $\mathbf{c}_{abs}^{t'} = \alpha^{t'}\mathbf{M}^{t'-1}$, where $\alpha^{t'} \in \mathbb{R}^K$ represents an attention distribution over the set of entries of memory $\mathbf{M}^{t'-1}$. Each scalar element $\alpha^{t',k}$ in an attention distribution $\alpha^{t'}$ can be calculated as: $\alpha^{t',k} = \texttt{softmax}(\texttt{cosine}(\mathbf{M}^{t'-1,k}, \mathbf{m}_a^{t'-1}))$, where we choose $\texttt{cosine}(\cdot, \cdot)$ as the align function, $\mathbf{M}^{t'-1,k}$ represents the $k$-th row memory vector, and $\mathbf{m}_a^{t'} \in \mathbb{R}^M$ is a add vector emitted by Hyper-RNN.

**Updating.** The Hyper-RNN can write and update abstract curriculum in the memory module. The write operation is performed as: $\mathbf{M}^{t'} = \mathbf{M}^{t'-1}(1 - \alpha^{t'}\mathbf{m}_e^{t'}) + \alpha^{t'}\mathbf{m}_a^{t'}$, where $\mathbf{m}_e^{t'} \in \mathbb{R}^M$ corresponds to the extent to which the current contents in the memory should be deleted.

Equipped with the above memory mechanism, the DRL learning algorithm can read the memory and utilize the retrieved information for policy learning. We incorporate the abstract curriculum into the value function by $Q(s^t, a^t, g^t, c_{abs}^{t'}; \phi_q)$. Similar to manually designed curricula, we minimize the Bellman error and define the loss function for the abstract curriculum as: $\mathcal{J}_{abstract} = \mathbb{E}_{(s^t, a^t, r^t, s^{t+1}, c_{abs}^{t'}) \sim H_{buf}}[(Q(s^t, a^t, c_{abs}^{t'}; \phi_q) - \dot{y})^2]$, where $\dot{y}$ is defined in Eqn. 1.

### 4.3 Bilevel Training of Hyper-RNN

After introducing the manually designed curricula in Sec. 4.1 and the abstract curriculum in Sec. 4.2, here we describe how we update the Hyper-RNN's parameters $\theta_h$, the parameters associated with the DRL agent $\phi_q$ and $\phi_\pi$. Since the Hyper-RNN's objective is to serve the DRL agent, we naturally formulate this task as a bilevel problem [56] of optimizing the parameters associated with multi-objective curricula generation by nesting one inner-level loop in an outer-level training loop.

**Outer-level training of Hyper-RNN.** Specifically, the inner-level loop for the DRL agent learning and the outer-level loop for training Hyper-RNN with hyper-gradients. The outer-level loss is defined as : $\mathcal{J}_{outer} = \mathcal{J}_{initial} + \mathcal{J}_{goal} + \mathcal{J}_{reward} + \mathcal{J}_{abs}$.

Since the manually designed curricula and abstract curricula are all defined in terms of $Q$-function, for the implementation simplicity, we combine them together $\mathcal{J}_{outer} = \mathbb{E}_{s^t, a^t, s^{t+1}, a^{t+1} \sim H_{buf}}[(Q(s^t, a^t, \mathbf{c}_{goal}, \mathbf{c}_{rew}, \mathbf{c}_{init}, \mathbf{c}_{abs}; \phi_q) - \dddot{y})^2]$. Following the formulation and implementation in [56], we obtain $\theta_h^* = \text{argmin}(\theta_h; \mathcal{J}_{outer}(\text{argmin}(\phi; \mathcal{J}_{inner}(\theta_h, \phi))))$.

**Inner-level training of DRL agent.** The parameters associated with the inner-level training, $\phi_q$ and $\phi_\pi$, can be updated based on any RL algorithm. In this paper, we use Proximal Policy Optimization (PPO) [63] which is a popular policy gradient algorithm that learns a stochastic policy.

## 5 Experiments

We evaluate and analyze our proposed MOC DRL on the CausalWorld [64], as this environment enables us to easily design and test different types of curricula in a fine-grained manner. This environment also provides wrapper makes the environment to execute actions on the real robot, which

can be used in sim2real experiments. It should be noted that we do not utilize any causal elements of the environment. It is straightforward to apply our method to other DRL environments without major modification. Moreover, the training and evaluation task distributions are handled by CausalWorld. Take task "Pushing" as an example: for each outer loop, we use CausalWorld to generate a task with randomly sampled new goal shapes from a goal shape family

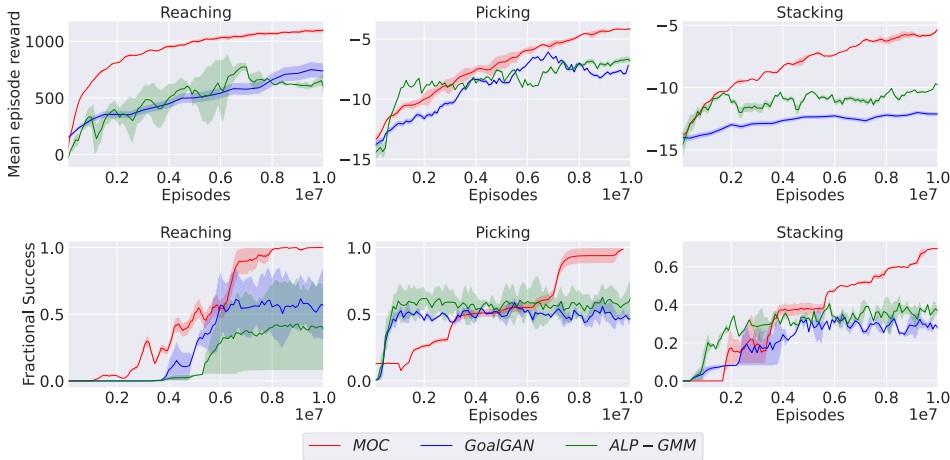

Figure 2: Comparisons with state-of-the-art ACL algorithms. Each learning curve is computed in three runs with different random seeds.

## 5.1  Comparing MOC with state-of-the-art ACL methods

We compare our proposed approach with the other state-of-the-art ACL methods: (1) GoalGAN [26], which uses a generative adversarial neural network (GAN) to propose tasks for the agent to finish; (2) ALP-GMM [5], which models the agent absolute learning progress with Gaussian mixture models. None of these baselines utilize multiple curricula.

Fig. 2 shows that MOC outperforms other ACL approaches in terms of mean episode reward, fractional success, and sample efficiency. Especially, MOC increases fractional success by up to 56.2% in all of three tasks, which illustrates the effectiveness of combining multiple curricula in a synergistic manner.

## 5.2  Ablation Study

Our proposed MOC framework consists of three key parts: the Hyper-RNN trained with hyper-gradients, multi-objective curriculum modules, and the abstract memory module. To get a better insight into MOC, we conduct an in-depth ablation study on probing these components. We first describe the MOC variants used in this section for comparison as follows: (1) $\textbf{MOC}_{Base-}$: MOC has the Hyper-RNN and the memory module but does not have the Base-RNNs for manually designed curricula. (2) $\textbf{MOC}_{Memory-}$: MOC has the Hyper-RNN to generate three curriculum modules but does not have the memory module. (3) $\textbf{MOC}_{Memory-,Hyper-}$: MOC has Base-RNNs but does not have memory and Hyper-RNN components. It independently generates manually designed curricula. (4) $\textbf{MOC}_{Memory-,Goal+}$: MOC with Hyper-RNN and one Base-RNN, but without the memory module. It only generates the subgoal curriculum as our pilot experiments show that it is consistently better than the other two manually designed curricula and is easier to analyze its behavior by visualizing the state visitation.

**Ablations of Hyper-RNN.** By comparing $\text{MOC}_{Memory-}$ with $\text{MOC}_{Memory-,Hyper-}$ as shown in Fig. 3, we can observe that letting a Hyper-RNN generate the parameters of different curriculum modules indeed helps in improving the sample efficiency and final performance. The advantage is even more obvious in the harder tasks `pick and place` and `stacking`. The poor performance of

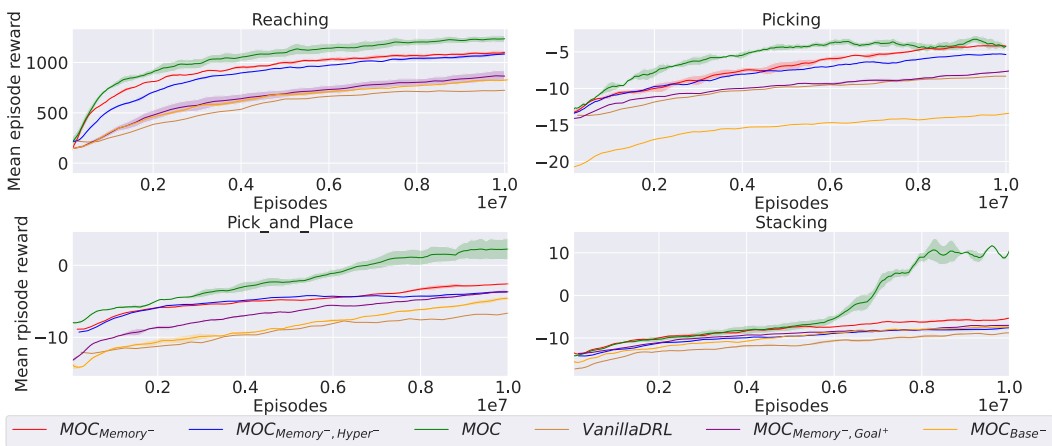

Figure 3: Comparison of algorithms with and without memory component on all four tasks. Each learning curve is obtained by three independent runs with different random seeds.

$\text{MOC}_{Memory^-, Hyper^-}$ may be caused by the potential conflicts among the designed curricula. For example, without coordination between the initial state curriculum and the goal curriculum, the initial state generator may set an initial state close to the goal state, which is easy to achieve by an agent but too trivial to provide useful training information to achieve the final goal. In sharp contrast, the Hyper-RNN can solve the potential conflicts from different curricula. All the curriculum modules are dynamically generated by the same hyper-net, and there exists an implicit information sharing between the initial state and the goal state curriculum generator. We put extra experimental results in Appendix Sec. A.3, Fig. 5.

**Ablations of the memory module.** We aim to provide an empirical justification for the use of the memory module and its associated abstract curriculum. By comparing MOC with $\text{MOC}_{Memory^-}$ as shown in Fig. 3, we can see that the memory module is crucial for MOC to improve sample efficiency and final performance. Noticeably, in `pick and place` and `stacking`, we see that MOC gains a significant improvement due to the incorporation of the abstract curriculum. We expect that the abstract curriculum could provide the agent with an extra implicit curriculum that is complementary to the manually designed curricula. We also find that it is better for the Hyper-RNN to learn the abstract curriculum while generating other manually designed curricula. Learning multiple manually designed curricula provides a natural curriculum for the Hyper-RNN itself since learning easier curriculum modules can be beneficial for learning of more difficult curriculum modules with parameters generated by the Hyper-RNN.

**Ablations of individual curricula.** We now investigate how gradually adding more curricula affects the training of DRL agents. By comparing $\text{MOC}_{Memory^-, Goal^+}$ and $\text{MOC}_{Memory^-}$ as shown in Fig. 3, we observe that training an agent with a single curriculum receives less environmental rewards as compared to the ones based on multiple curricula. This suggests that the set of generated curricula indeed helps the agent to reach intermediate states that are aligned with each other and also guides the agent to the final goal state.

## 6 Limitations

This paper presents a multi-objective curricula learning approach for solving challenging deep robotics tasks. However, it may be interesting to improve the sample efficiency of policy learning. Our method utilizes bi-level optimization, which elongates the training efficiency in terms of interactions between robots and environments. We argue that this limitation could be largely alleviated by replacing the model-free policy learning method with the model-based policy learning method.

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
