# OpenReview forum: "Learning Multi-Objective Curricula for Robotic Policy Learning"
_robot-learning.org/CoRL/2022/Conference — CoRL 2022 Poster_

### Official Review · Reviewer_oFwa · 2022-07-17

**Originality:** Good
**Technical Quality:** Very Good
**Clarity Of Presentation:** Good
**Impact:** 3

**Recommendation:**

Weak Accept: I recommend accepting the paper, but will not argue for my recommendation if the majority of other reviewers have a different opinion.

**Summary:**

This paper proposes a method to generate manually designed curricula as well as automatically generated abstract curricula for policy learning in the reinforcement learning setting. These curricula influence the data collection of the agent and thus can gradually help the agent to solve the task. By this, the agent's sample efficiency improves and it is able to solve the task with qualitative policies. The proposed curricula are coordinated via a multi-task hyper-net which parameterizes the curriculum base networks. The method is benchmarked against other automatic curriculum learning approaches on six robotic manipulation tasks.

**Issues:**

My main concerns were already described in the 'Strengths and Weaknesses' section. I would be happy if the issues would be considered during the rebuttal.

**Quality Of The Limitations Section:**

Limitations are addressed clearly

**Reviewer Expertise:**

2: The reviewer is willing to defend the evaluation, but it is quite likely that the reviewer did not understand central parts of the paper

**Robotics Focus:**

Highly relevant to robotics but no hardware experiments

**Strengths And Weaknesses:**

The paper is well written such that the reader can easily follow the intuition and the ideas. It shows a thorough experiments part including a thorough ablation study.

Even though some related work was mentioned in the main text, I was missing an explicitly related work part while reading. In my opinion, summarizing the related work in a chapter and explicitly mentioning the differences between the proposed method to those is useful for the reader to be able to have an idea of the novelty. There are also a couple of other works which I could find on the internet which consider automatically adjusting their curriculum in the context of reinforcement learning and might be worth a discussion:
- https://proceedings.neurips.cc/paper/2020/file/68a9750337a418a86fe06c1991a1d64c-Paper.pdf
- http://proceedings.mlr.press/v100/klink20a/klink20a.pdf
- https://proceedings.mlr.press/v164/celik22a.html

**Summary Of Recommendation:**

I like the proposed idea in this paper and I believe it is well written such that the reader can easily understand the purpose of the proposed method. The experiments show the advantages gained through the method. I would have appreciated an explicit related work part. However, I am willing to accept this paper.

---

> ### Author Response · Authors · 2022-08-20
> **Response to Reviewer oFwa**
>
> Thanks for your constructive suggestions and here are our responses to your questions.
>
> We’ve put the related work from the appendix to the main text lines 68-102 in the color sky blue. The suggested related work is discussed in this revision in lines 75-80 in the color red violet.

---

> > ### Comment · Reviewer_oFwa · 2022-08-25
> > **Response to the authors**
> >
> > I thank the reviewers and stay with my opinion to accept the paper.

---

### Official Review · Reviewer_9eoj · 2022-07-29

**Originality:** Very Good
**Technical Quality:** Very Good
**Clarity Of Presentation:** Good
**Impact:** 4

**Recommendation:**

Strong Accept: I recommend accepting the paper and will argue for my recommendation even if other reviewers hold a different opinion.

**Summary:**

This paper proposes an architecture and learning pipeline to generate multi-objective curricula.

The basic ideas are:
1. Using a hypernet to generate the weights of models that further generate each manually designed curricula. This includes learnable initial states, subgoals, and shaped reward functions.
2. To enable futher flexibility, the authors propose to use an augmented memory to support the learning agent, which can be written by the hypernetwork. So this is a general form of "curricula" besides the aforementioned manually designed curricula.
3. Frame ACL as a bilevel optimization problem and optimize hypernet and $Q$ and $\pi$ jointly.

**Issues:**

1. I do not see an obvious reason why using $\gamma$ for goal and not for the discount factor.
2. Line 145, For the shaping reward, why learning $\mu$ and not fixing it to the discount factor $\lambda$ (in your notation). If it is not $\lambda$, then in theory it cannot guarantee optimality anymore.
3. From the presentation perspective, I think the authors can explicitly list their contributions at the end of introduction using bullet points, which can potentially help others to quickly get the novelties proposed in this work.

**Quality Of The Limitations Section:**

Limitations are addressed clearly

**Reviewer Expertise:**

3: The reviewer is fairly confident that the evaluation is correct

**Robotics Focus:**

Relevant but unlikely to deploy to hardware in near future

**Strengths And Weaknesses:**

Strengths:
1. I think this paper proposes a novel problem (multi-objective curricula), a general perspective (bilevel optimization for ACL), and a novel architecture (hypernet with write-only memory as a general curricula).
2. The main ideas are straightforward and reasonable.
3. Experiment results are convincing. Particularly, it has been shown that the combination of hypernet and augmented memory is necessary.

Weakness:
I find this work very easy to follow and no significant weakness from my opinion.




**Summary Of Recommendation:**

I think there are several novel points this paper brings:
1. A bilevel optimization (BO) view of ACL, though not completely novel, but to my knowledge this is the first paper explicitly using BO to train ACL.
2. Multi-objective curriculum via hypernetwork is an interesting proposal, which automatically connects all curricula.
3. The augmented memory is a novel form of "curricula" that is very flexible.

---

> ### Author Response · Authors · 2022-08-20
> **Response to Reviewer 9eoj**
>
> Thanks for your constructive suggestions and here are our responses to your questions. We also revised our paper according to your comments, which are shown in color teal.
>
> ### Q1
>
> > I do not see an obvious reason why using γ for goal and not for the discount factor.
> >
>
> R: Thank you for this valuable comment, we’ve changed this notation in the revised version in color teal.
>
> ### Q2
>
> > Line 145, For the shaping reward, why learning μ and not fixing it to the discount factor λ (in your notation). If it is not λ, then in theory it cannot guarantee optimality anymore.
> >
>
> R: The hyper-parameter $\mu$ is a fixed constant value, which needs to be tuned manually. We do not learn this parameter but manually tweak this value.
>
> ### Q3
>
> > From the presentation perspective, I think the authors can explicitly list their contributions at the end of introduction using bullet points, which can potentially help others to quickly get the novelties proposed in this work.
> >
>
> R: Thank you for the valuable comment. We’ve added explicit contributions to our revised paper lines 58-67 in the color sky blue.

---

### Official Review · Reviewer_HfsD · 2022-08-03

**Originality:** Good
**Technical Quality:** Good
**Clarity Of Presentation:** Good
**Impact:** 3

**Recommendation:**

Weak Reject: I recommend rejecting the paper, but will not argue for my recommendation if the majority of other reviewers have a different opinion.

**Summary:**

The paper proposes a learning architecture that guides the learning process of an agent by simultaneously: 1) selecting an initial state from the agent starts the next episode; 2) selecting the agent's goal; and 3) selecting the reward function. Experiments on simulated robot environments (reaching, pick and place, and stacking) show some improvements relative to a few select baseline approaches.

**Issues:**



Questions for authors:

For the inputs to the Hyper-RNN, can you elaborate more on why you only consider the final state of the last episode?

Question on the 3 environments: these problems appear difficult mostly because the environmental reward function is sparse, in other words, it is engineered very poorly. Have you considered settings where some thought was put into the reward function to make the task easier to learn? Also, when each new episode is generated, what is randomized (e.g., positions of objects) and what remains the constant (e.g., the goal)?

In Figure 2, are the episodes likely to have more or less the same number of time steps? is it possible that some approaches result in episodes that are much shorter or longer? it may be good to plot the performance in terms of number of time steps as opposed to number of episodes as ultimately it is the interaction with the environment that we want to minimize

What were the fundamental limitations that prevent the baseline approaches from learning to complete the task all the time? theoretically, they should be able to do so given enough data but it looks like they've converged before hitting high success rates. Also, are the baselines essentially learning from scratch without a curricula? Why not have a simply DNN baseline that just tries to learn from scratch

In the limitations, cam you elaborate on what you mean by "elongates the training efficiency"?


Comments on parts of the paper:

In the last sentence of the 1st paragraph of introduction, more broadly, curriculum learning can also be used to modify the world itself, e.g., add new entities to the world, change the behaviors of other agents, etc., see:

Narvekar, Sanmit, et al. "Source task creation for curriculum learning." Proceedings of the 2016 international conference on autonomous agents & multiagent systems. 2016.

Somewhere in the introduction, you should elaborate what you mean by output a curriculum? how is curriculum represented? Sometimes a curriculum is a sequence of tasks, sometimes a graph of tasks, sometimes it is just an ordering of data points.

A running illustrative example would really help is space permits

In preliminaries, the RL section should include the notion of goals since goals are used in this work

In the second paragraph of Section 2, "the output of D is a curriculum such as an initial state.", that's actually false. An episode with some particular initial state is part of a bigger curriculum, yes, but it is not in itself the curriculum. The curriculum is instead, the sequence of generated episodes, each with different initial state, goal, etc.

In Figure 3, shouldn't all methods do the same at the very beginning with no training?




**Quality Of The Limitations Section:**

Additional details required

**Reviewer Expertise:**

4: The reviewer is confident but not absolutely certain that the evaluation is correct

**Robotics Focus:**

Relevant but unlikely to deploy to hardware in near future

**Strengths And Weaknesses:**


Strengths:

The approach is novel and the idea of learning how to initialize an episode to guide the agent's learning is creative.

Ablation study is informative

Weaknesses:

There are a few clarity issues with the paper's writing and presentations -- see detailed comments under 'issues'

There needs to be better justification for why these particular baselines make sense -- see questions in detailed comments



**Summary Of Recommendation:**

The recommendation is based on the strengths and weaknesses outlined in the review, as well as the list of issues. The recommendation is subject to change depending on the authors' responses.

---

> ### Author Response · Authors · 2022-08-20
> **Response to Reviewer HfsD (3)**
>
> ### Q9
>
> > In the second paragraph of Section 2, "the output of D is a curriculum such as an initial state.", that's actually false. An episode with some particular initial state is part of a bigger curriculum, yes, but it is not in itself the curriculum. The curriculum is instead, the sequence of generated episodes, each with different initial state, goal, etc.
> >
>
> R: Thank you for pointing this out. Yes, more precisely, the curriculum should be a sequence of generated episodes. Each episode with different desired goals, initial states, rewards, etc. We’ve revised this sentence in lines 120-121 in the color magenta.
>
> ### Q10
>
> > Q: In Figure 3, shouldn't all methods do the same at the very beginning with no training?
> >
>
> R: All the curves are plotted with the same logging frequency (i.e. 10,000 episodes). The initial dot actually reports the result after training 10,000 episodes, which result in various performance at the beginning. This type of figure can also be seen in [8] Figure 5 and in [9] Figure 2.
>
> [8] “Reinforcement Learning with Deep Energy-Based Policies” ICML 2017. [http://proceedings.mlr.press/v70/haarnoja17a/haarnoja17a.pdf](http://proceedings.mlr.press/v70/haarnoja17a/haarnoja17a.pdf)
>
> [9] “Continuous Deep Q-Learning with Model-based Acceleration”. ICML 2016. [http://proceedings.mlr.press/v48/gu16.pdf](http://proceedings.mlr.press/v48/gu16.pdf)

---

> ### Author Response · Authors · 2022-08-20
> **Response to Reviewer HfsD (2)**
>
> ### Q4
>
> > Q: What were the fundamental limitations that prevent the baseline approaches from learning to complete the task all the time? theoretically, they should be able to do so given enough data but it looks like they've converged before hitting high success rates. Also, are the baselines essentially learning from scratch without a curricula? Why not have a simply DNN baseline that just tries to learn from scratch
> >
>
> R: The reason we choose CausalWorld as the evaluation experiment is that it is hard to train a DNN baseline from scratch without curricula (reference CausalWorld). This DNN baseline can be found on page 7 Figure 3 **vanillaDRL,** which is also summarized in the table below. Moreover, it is also hard for baseline methods to train a good policy with one curriculum only (i.e. goal curriculum for GoalGAN and ALP-GMM). Theoretically, with proper exploration and exploitation tradeoff, as well as certain conditions on value/policy representation class, an RL algorithm could potentially converge. However, in practice, there are several difficulties in training the optimal policy into convergence. 1) Exploration vs. Exploitation: the reason that baseline algorithms converge to sub-optimal policies is that the current goal curriculum cannot provide a good policy exploration for an agent to collect more rewards. In this case, the agent tends to exploit the existing sub-optimal policy. 2) With longer training episodes, it is possible that the baseline algorithms can explore a policy that outperforms the old one, and result in reaching the optimal policy. But, it takes longer episodes to train these baselines, which also raises the training efficiency limitations of these baselines.
>
> | Algorithms  |    Reaching     |    Picking    |    Stacking   |
> |:-----------:|:---------------:|:-------------:|:-------------:|
> |   ALP-GMM   |   690.84+/-179  |  -6.3+/-1.96  |  -9.76+/-0.59 |
> |   Goal-GAN  |   702.94+/-242  |  -6.42+/-1.86 | -11.54+/-0.52 |
> | Vanilla DRL | 557.86+/-168.26 | -10.28+/-1.59 | -12.07+/-1.87 |
>
> ### Q5
>
> > In the limitations, cam you elaborate on what you mean by "elongates the training efficiency"?
> >
>
> R: In this paper, we utilize the bi-level optimization method, which is one type of meta-learning method, to extract the common knowledge across tasks. As illustrated in [5,6], the bi-level optimization method shows good performance but suffers from the training efficiency. Because bi-level optimization requires storing all the intermediate computation graphs for each step of gradient descent.
>
> [5] “Gradient-based Hyperparameter Optimization through Reversible Learning” ICML 2015. [https://arxiv.org/pdf/1502.03492.pdf](https://arxiv.org/pdf/1502.03492.pdf)
>
> [6] “HyperShot: Few-Shot Learning by Kernel HyperNetworks.” [https://arxiv.org/pdf/2203.11378.pdf](https://arxiv.org/pdf/2203.11378.pdf)
>
> ### Q6
>
> > In the last sentence of the 1st paragraph of introduction, more broadly, curriculum learning can also be used to modify the world itself, e.g., add new entities to the world, change the behaviors of other agents, etc., see: Narvekar, Sanmit, et al. "Source task creation for curriculum learning." Proceedings of the 2016 international conference on autonomous agents & multiagent systems. 2016.
> >
>
> R: Thank you for the valuable comment, we’ve added this sentence in the revised version lines 23-25 in the color magenta.
>
> ### Q7
>
> > Somewhere in the introduction, you should elaborate what you mean by output a curriculum? how is curriculum represented? Sometimes a curriculum is a sequence of tasks, sometimes a graph of tasks, sometimes it is just an ordering of data points.
> >
>
> R: In our paper, we follow the ideal of automatic goal generation [7] and extended this idea to generate multiple curricula. In particular, the curriculum refers to a sequence of tasks with different desired goals, rewards, initial states, etc. We’ve revised this sentence in lines 34-35 in the color magenta.
>
> [7] “Automatic Goal Generation for Reinforcement Learning Agents.” IJCAI. *ArXiv* abs/1705.06366 (2018): n. pag.
>
> ### Q8
>
> > In preliminaries, the RL section should include the notion of goals since goals are used in this work
> >
>
> R: Thank you for the valuable comment, we’ve added this notation in the revised version lines 106-108 in the color magenta.

---

> ### Author Response · Authors · 2022-08-20
> **Response to Reviewer HfsD (1)**
>
> Thanks for your constructive suggestions and here are our responses to your questions. We also revised our paper according to your comments, which are shown in the magenta text.
>
> ### Q1
>
> > Q: For the inputs to the Hyper-RNN, can you elaborate more on why you only consider the final state of the last episode?
> >
>
> R: The motivations for using the final state of the last episode can be summarized in the followings:
>
> ### Chain the gradients
>
> We aim to link the hyper-RNN parameter training to policy learning so that we can optimize the hyper-RNN parameters within the meta-learning framework [1]. Specifically, the parameters of the actor-network and critic-network are optimized in the inner loop; The parameters of the hyper-RNN are optimized in the outer loop. By adding the final state to the hyper-RNN, we are able to chain the final loss gradients through both the outer and inner loops to optimize the hyper-RNN, actor-network, and critic-network.
>
> ### Why only the last episode
> Second, similar to [2], we can treat the one episode of hyper-RNN generation as a contextual bandit problem. In the classic multi-armed bandit problem, a gamer iteratively plays according to the observed rewards. The contextual bandit algorithm is an extension of the multi-armed bandit approach where we factor in the agent’s current context when choosing a bandit (generating the parameters for base-RNN). The context affects how a reward is associated with each bandit, so as contexts change, the model should learn to adapt its bandit choice. In our experiment, we mainly focus on a goal-conditioned problem. We treat the final state of the last episode as our current context as it represents whether we reach the goal. The hyper-RNN observes the current context (final state of the last episode) and takes actions (by generating parameters for base-RNN) that receive more rewards. In this case, the last episode suffices to provide enough training signals for the hyper-RNN and works well in practice. We’ve added this explanation in the revised paper on page 4 footnote 2 in the color magenta.
>
> [1] Generalized Inner Loop Meta-Learning. [https://arxiv.org/pdf/1910.01727.pdf](https://arxiv.org/pdf/1910.01727.pdf)
>
> [2] Been There, Done That: Meta-Learning with Episodic Recall. ICML 2018. [https://arxiv.org/pdf/1805.09692.pdf](https://arxiv.org/pdf/1805.09692.pdf)
>
> ### Q2
>
> > Q: Question on the 3 environments: these problems appear difficult mostly because the environmental reward function is sparse, in other words, it is engineered very poorly. Have you considered settings where some thought was put into the reward function to make the task easier to learn? Also, when each new episode is generated, what is randomized (e.g., positions of objects) and what remains the constant (e.g., the goal)?
> >
>
> R: All the environments can provide dense reward signals as well as sparse rewards.
> In each episode, the CausalWorld environment generates a fixed characteristic goal shape of this distribution with fixed initial building block poses. The curriculum generator generates subgoals, initial states, and shaping rewards in each episode.
>
> ### Q3
>
> > Q: In Figure 2, are the episodes likely to have more or less the same number of time steps? is it possible that some approaches result in episodes that are much shorter or longer? it may be good to plot the performance in terms of number of time steps as opposed to number of episodes as ultimately it is the interaction with the environment that we want to minimize
> >
>
> R: For clearness and fairness, we intentionally train all the comparison schemes in the same timesteps for all different tasks. As we can see in Figure2, as well as appendix Figure 10, different algorithms show various convergence rates or performances in these five tasks. For example, in “reaching” and “picking”, MOC shows convergence after around 0.65*e7 and 0.8*e7 episodes, respectively. The same phenomenon can also be discovered in other tasks and baselines. This training manner can be also seen in [3] Figure2 and in [4] Figure 5.
>
> [3] Guided Policy Search. ICML 2013. [http://proceedings.mlr.press/v28/levine13.pdf](http://proceedings.mlr.press/v28/levine13.pdf)
>
> [4] Model-Agnostic Meta-Learning for Fast Adaptation of Deep Networks. ICML 2017. [http://proceedings.mlr.press/v70/finn17a/finn17a.pdf](http://proceedings.mlr.press/v70/finn17a/finn17a.pdf)

---

### Official Review · Reviewer_g3te · 2022-08-06

**Originality:** Good
**Technical Quality:** Very Good
**Clarity Of Presentation:** Fair
**Impact:** 3

**Recommendation:**

Weak Accept: I recommend accepting the paper, but will not argue for my recommendation if the majority of other reviewers have a different opinion.

**Summary:**

This paper presents an automatic curriculum learning framework for multi-objective curricula (MOC). The key insight is to use a multitask hyper-net module on top of multiple curricula generating neural networks. The hyper-net module resolves potential conflicting gradients between the lower level modules and parameterizes them accordingly. Each of these atomic module represents a single way of introducing curricula in RL: (i) subgoal sequence, (ii) initial states and (iii) reward function. Additionally, the external memory of the hyper-net is supposed to generate an “abstract“ curricula that is different from the predefined curricula above. The parameters of the abstract curriculum are added to the value function.



**Issues:**

The paper seems to be written in a rush and needs to be revised to improve clarity and flow of reading. Moreover, the contributions need to be explicitly formulated.

**Quality Of The Limitations Section:**

Limitations are addressed clearly

**Reviewer Expertise:**

2: The reviewer is willing to defend the evaluation, but it is quite likely that the reviewer did not understand central parts of the paper

**Robotics Focus:**

Highly relevant to robotics but no hardware experiments

**Strengths And Weaknesses:**

# Strengths
- The idea to learn individual curricula modules from scratch is interesting. It seems clever to learn distinct curricula and how to combine them at the same time to achieve good overall performance.
- Evaluations: MOC has been tested against state-of-the-art automatic curriculum learning approaches and seems to outperform them. Moreover, it is much appreciated that the authors perform a detailed ablation study.

# Weaknesses
- The paper is at parts hard to understand. For instance, it is not clear what the pros and cons of read and write-only memory are (discussion at the end of the introduction) at this part of the paper. The authors describe how their approach compares to previous work, but leave the reader with no intuition why restricting memory access is important at all. Another example is the last sentence of the automatic curriculum learning section, where it is not clear what the “history“ is.
- What exactly is this paper doing for the first time? Is it the introduction of the abstract curriculum module, or the combination of different types of curricula and their parallel tuning? The empirical evaluations are definitely also a contribution. A paragraph (that goes beyond the current contributions paragraph) saying explicitly how this paper differs from others would be helpful.

**Summary Of Recommendation:**

Despite that the paper is at parts not clearly written, the techniques introduced and the evaluations seem convincing. Also, this paper tackles an interesting and important research direction with potentially high impact.

---

> ### Author Response · Authors · 2022-08-20
> **Response to Reviewer g3te**
>
> Thanks for your constructive suggestions and here are our responses to your questions. We also revised our paper according to your comments, shown in sky blue text.
> ### Q1
>
> > Q: The paper is at parts hard to understand. For instance, it is not clear what the pros and cons of read and write-only memory are (discussion at the end of the introduction) at this part of the paper. The authors describe how their approach compares to previous work, but leave the reader with no intuition why restricting memory access is important at all. Another example is the last sentence of the automatic curriculum learning section, where it is not clear what the “history“ is.
> >
>
> ### Memory Design Pros and Cons
>
> **Pros**
>
> 1) Our motivation for proposing a memory module comes from the Neural Turing Machine [1]. The memory module is to mitigate the issue of multi-curricula interference. As shown in Figure 3 and discussed in lines 298-299, “Noticeably, in pick and place and stacking, we see that MOC gains a significant improvement due to the incorporation of the abstract curriculum. We expect that the abstract curriculum could provide the agent with an extra implicit curriculum that is complementary to the manually designed curricula.”
>
> 2) As we discussed in Line 47, the hyper-net can write entries to and update items in the memory, through which the DRL agent can interact with the environment under the **guidance** of the abstract curriculum maintained in the memory.
>
> 3) Hyper-RNN learns how to generate a series of progressive goals and initial states for the Base-RNN, **across multiple inner loops**, and the memory mechanism of the Hyper-RNN can **maintain** such information.
>
> **Cons**
>
> The cons are that the read and write operations may cause extra computation time and memory resources. But in our robotic learning settings, the extra computation time and memory resources are ignorable. Concretely, it incurs “linearly w.r.t the size of memory but gets a ~20 times speed up” [2]. Furthermore, the memory module is not the computation bottleneck in the proposed framework, which is acceptable in our design.
>
> ### Read and write-only Design Pros and cons
>
> **Pros**
>
> 1) The read and write-only memory improves performance and reduces system complexity compared to read and write memory design.
>
> We’ve also conducted additional experiments to justify this design. Instead of read and write-only memory, we directly predict the abstract curriculum information. As can be seen below, this classic memory is somewhat useful, but MOC still outperforms it. One possible reason might be that when the agent reads messages from the memory, the agent will attend to the vector, which is different from the classic memory (with read and write). This is explained in Sec. 3.2.
>
> |  Algorithms   |    Reaching     |    Picking    |    Stacking   | Pick and Place |   Pushing   |
> |:-------------:|:---------------:|:-------------:|:-------------:|:--------------:|:-----------:|
> | $MOC_{base-}$ | 596.02+/-178.90 | -15.56+/-1.79 | -10.09+/-2.36 |   -8.68+-2.27  | -8.68+-2.27 |
> | Classic Memory | 543.68+/-239.71 | -17.99+/-2.15 | -15.73+/-5.07 |  -13.04+-2.00  | -9.69+-3.69 |
>
> 2) This design prevents the agent from writing its own information into the memory module, which can interfere with the generated curriculum.
>
> **Cons**
>
> Theoretically, with a huge read-and-write memory module, the hyper-RNN can have the luxury to store information other than curriculum into the memory without interfering with the generated curriculum. But in practice, since the memory module cannot be too big, this may not be possible.
>
> ### History
>
> R: The history $H$ can consist of any information about past interactions, for example, in our experiments the history consists of the last state of the episode. We’ve revised this sentence in lines 119-120 in the color sky blue.
>
> [1] Neural Turing Machines. CoRR abs/1410.5401. [https://arxiv.org/abs/1410.5401](https://arxiv.org/abs/1410.5401).
>
> [2] Memory Augmented Policy Optimization for Program Synthesis and Semantic Parsing. Nips 2018. [https://proceedings.neurips.cc/paper/2018/file/f4e369c0a468d3aeeda0593ba90b5e55-Paper.pdf](https://proceedings.neurips.cc/paper/2018/file/f4e369c0a468d3aeeda0593ba90b5e55-Paper.pdf)
>
> ### Q2
>
> > Q: What exactly is this paper doing for the first time? Is it the introduction of the abstract curriculum module, or the combination of different types of curricula and their parallel tuning? The empirical evaluations are definitely also a contribution. A paragraph saying explicitly how this paper differs from others would be helpful.
> >
>
> R: The answers about related work are posted in the comments above see main text lines 68-102.
>
> ### Q3
>
> > Q: The paper seems to be written in a rush and needs to be revised to improve clarity and flow of reading. Moreover, the contributions need to be explicitly formulated.
> >
>
> R: The answers about contributions are posted in the comments above see revised paper lines 58-67.

---

> > ### Comment · Reviewer_g3te · 2022-08-25
> > **Response to the authors**
> >
> > Thank you for clarifying my questions. I will keep my score at “weak accept”.

---

### Meta-Review · Area_Chair_L2dH · 2022-08-13

**Recommendation:** Accept (Poster)
**Confidence:** 4

**Metareview:**


# Strengths
- The proposed approach is novel and the idea of combining individual curricula focused on different aspects such as initial states or reward functions is interesting and promising
- The evaluation of the approach was well done and included comparisons to SOTA methods and detailed ablations

# Weaknesses
- The paper requires some additional clarifications as pointed out by the reviewers
- An dedicated related work section would further improve clarity
- The reviewers highlighted some additional related work that should be addressed in the paper.

# Post-Rebuttal Update
The additional related work section and the added clarifications definitely improve the presentation of the work and help to position the paper amongst prior work.
While additional baselines and more complex experiments/scenarios, ideally in real-world settings, would further improve the paper, I do believe the paper should be published.
I encourage the reviewer to polish the text once more. There are a few spelling/grammar mistakes, e.g., in the experiment section.
Furthermore, the paper should be shortened back to 8 pages for the camera-ready version.


**Best Paper Nomination:**

No